# Acute Macular Neuroretinopathy after SARS-CoV-2 Infection: An Analysis of Clinical and Multimodal Imaging Characteristics

**DOI:** 10.3390/diagnostics13243600

**Published:** 2023-12-05

**Authors:** Jiyang Tang, Siying Li, Zongyi Wang, Ye Tao, Linqi Zhang, Hong Yin, Heng Miao, Yaoyao Sun, Jinfeng Qu

**Affiliations:** 1Department of Ophthalmology, Peking University People’s Hospital, Beijing 100044, China; 2Eye Diseases and Optometry Institute, Beijing 100044, China; 3Beijing Key Laboratory of Diagnosis and Therapy of Retinal and Choroid Diseases, Beijing 100044, China; 4College of Optometry, Peking University Health Science Center, Beijing 100044, China

**Keywords:** acute macular neuroretinopathy, SARS-CoV-2, multimodal imaging

## Abstract

Background: This study aimed to analyze clinical and multimodal imaging characteristics of acute macular neuroretinopathy (AMN) post-recent severe acute respiratory syndrome coronavirus 2 (SARS-CoV-2) infection. Methods: Retrospective observational study. Medical records and multimodal imaging of 12 AMN eyes of eight patients (six female and two male) with recent SARS-CoV-2 infection were retrospectively analyzed. Results: Four patients (50%) presented with bilateral AMN. Fundus ophthalmoscopy revealed a reddish-brown lesion around the macula, and two eyes had cotton-wool spots at the posterior pole. Three eyes showed mild hypo-autofluorescence. All FFA images (7 eyes) showed no abnormal signs. On OCT scans, all eyes showed outer nuclear layer (ONL) thinning, 8 eyes (66.7%) showed ONL hyperreflectivity, 5 eyes (41.7%) showed outer plexiform layer (OPL) hyperreflectivity, 8 eyes (66.7%) showed interdigitation zone (IZ) disruption, 11 eyes (91.6%) showed ellipsoid zone (EZ) disruption, 2 eyes (16.7%) showed cotton-wool spots and inner plexiform layer (IPL) hyperreflectivity, 1 eye (8.3%) had intraretinal cyst and 1 eye (8.3%) had inner nuclear layer (INL) thinning. Persistent scotoma, ONL hyperreflectivity and IZ/EZ disruption as well as recovery of OPL hyperreflectivity were reported after follow-up in three cases. Conclusions: AMN post-SARS-CoV-2 mostly affected young females and could present unilaterally or bilaterally. Dark lesions on IR reflectance and outer retinal hyperreflectivity on OCT are useful in diagnosing AMN. OPL/ONL hyperreflectivity on OCT could disappear after follow-up, but ONL thinning and IZ/EZ could persist.

## 1. Introduction

Acute macular neuroretinopathy (AMN) is a rare retinal disorder first reported by Bos and Deutman in 1975 [1]. AMN, more often affecting young females, could present unilaterally or bilaterally with symptoms including scotoma and photopsia, with or without decreased vision. Characteristic fundus appearance shows isolated or multiple reddish-brown, wedge-shaped lesions around the fovea [2]. Common reported risk factors include nonspecific flulike illness, fever and oral contraceptives; other risk factors including exposure to epinephrine, ephedrine and systemic shock were also reported to be related to AMN [2]. The exact etiology of AMN remains unclear. Studies using en face optical coherence tomography (OCT) and spectral domain (SD) OCT suggest that the pathogenesis of AMN may be related to retinal deep capillary ischemia [3,4] as well as choriocapillaris involvement [5].

Severe acute respiratory syndrome coronavirus 2 (SARS-CoV-2) infection has been reported to cause vascular endothelial cell damage and thromboinflammation, resulting in microthrombi and microvascular dysfunction [6,7]. Following the pandemic of SARS-CoV-2, post-SARS-CoV-2 infection or vaccination, retinal vascular complications including retinal artery occlusion (RAO), retinal vein occlusion (RVO), acute paramacular central medial retinopathy (paracentral acute middle maculopathy, PAMM) and AMN have been reported [8].

This study aimed to provide more insights into the clinical diagnosis and treatment of AMN as well as its association with SARS-CoV-2 by retrospectively analyzing the multimodal imaging features of AMN eyes in patients with a history of recent SARS-CoV-2 infection at our center.

## 2. Materials and Methods

This was a retrospective observational study. Medical records of eight patients (12 eyes) diagnosed with AMN with recent SARS-CoV-2 infection at Peking University People’s Hospital between December 2022 and February 2023 were retrospectively reviewed. This study was conducted in accordance with the Declaration of Helsinki for research involving human subjects.

Demographic and relevant clinical findings available at each visit were collected and analyzed by the authors. Patients underwent a complete ophthalmic examination, including best-corrected visual acuity (BCVA) using decimal charts, slit-lamp biomicroscopy, tonometry and indirect fundus ophthalmoscopy. Fundus photography (FP), fundus autofluorescence (AF) and fluorescein angiography (FA) were performed using an Optos 200Tx (Optos plc, Dunfermline, Scotland, UK). Infrared (IR) reflectance was performed using a Spectralis HRA OCT (Heidelberg Engineering, Heidelberg, Germany) or a BM-400 K BMizar OCT (TowardPi Medical Technology, Beijing, China). OCT and OCTA were performed using a CIRRUS HD-OCT 5000 AngioPlex (Zeiss Meditec, Inc., Dublin, CA, USA, software version 9.6.1.57321), an Avanti RTVue XR 100-2 AngioVue (Optovue, Fremont, CA, USA, software version 2017.1.0.155) or a BM-400K BMizar OCT (TowardPi Medical Technology, Beijing, China, software version 1.2.10.1). OCTA was performed on selected patients with a scanning area of 6 × 6 mm or 12 × 12 mm, centered on the fovea. The automatic segmentation provided by the OCT-A software was manually adjusted by retina specialists for correct visualization of the superficial retinal capillary plexus (SCP), the deep retinal capillary plexus (DCP), the avascular outer retina slab and the choriocapillaris. Multifocal electroretinography (mfERG) was performed using a Color Ganzfeld Q450 (Roland Consult, Bradenburg, Germany). SITA-FAST visual field tests were performed using a Humphrey 750 i (Zeiss Meditec, Inc., Dublin, CA, USA). Inclusion criteria included the following: patient with a clinical diagnosis of AMN; and recent infection of SARS-CoV-2 prior to AMN-related symptoms. Diagnosis of AMN was based on history of scotoma corresponding to hyporeflectivity on IR reflectance and ONL hyperreflectivity on OCT scans. Evidence of SARS-CoV-2 infection was based on positive results of a PCR test or antigen rapid test with nasopharyngeal swab. Exclusion criteria included the following: coexisting retinal or choroidal disorders; and flulike symptoms unrelated to SARS-CoV-2 infection prior to AMN-related symptoms.

## 3. Results

Eight AMN patients (with 12 affected eyes) were included in this study, six of whom were female. Four patients presented unilaterally while the other four presented bilaterally. Patients were aged 30 ± 11 (16~56, median 28) years. Six patients complained of scotoma, four complained of blurred vision and two complained of photopsia, while one patient presented with no symptoms. All patients had a recent history of SARS-CoV-2 infection and high fever (>39 °C). Out of the seven patients who presented with symptoms, time from SARS-CoV-2-related fever to onset of ophthalmic symptoms was 1~10 days (median 2 days). All patients received NSAIDs for COVID-19-induced fever and none reported use of anticoagulants. All patients denied any history of oral contraceptive use. All but one patient (case 8) denied use of glucocorticoids. Case 8 received regular oral glucocorticoids for her comorbidity of systemic lupus erythematosus (SLE).

The BCVA of the affected eyes was 0.23 ± 0.30 (0.00~0.92, median 0.05). Anterior segment examination was unremarkable in all patients. Fundus examination of all affected eyes showed reddish-brown lesions in the macular area (Figure 1A,J, Figure 2A, Figure 3A,G and Figure 4A,F), and one patient had cotton-wool spots at the posterior pole of both eyes (Figure 5A,B). Among the eight affected eyes that underwent IR examination, dark, well-demarcated areas of the corresponding lesions could be seen on the IR images of seven affected eyes (Figure 1G,P, Figure 2F, Figure 3D,J and Figure 6E), and the multiple lesions could be observed in a “petaloid” pattern (Figure 1G and Figure 3D,J). Slightly decreased autofluorescence was seen on AF images of both eyes (Figure 3A,H) in case 3 and the left eye of case 4 (Figure 4G). FFA examinations of four patients revealed no vascular abnormalities (Figure 1C,L, Figure 2C, Figure 3C,I and Figure 4C,J). Of the three patients (and three affected eyes) that underwent visual field examination, all three eyes showed paracentral or central scotoma consistent with the patient’s visual symptoms (Figure 1F,O and Figure 2I).

Characteristics of OCT scans of all affected eyes are shown in Table 1. OCT of all 12 affected eyes showed changes in the outer retinal structure; all affected eyes showed outer nuclear layer (ONL) thinning; 8 eyes (66.7%) showed hyperreflectivity in the ONL; 5 eyes (41.7%) showed hyperreflectivity in the outer plexiform layer (OPL); 8 eyes (66.7%) showed interdigitation zone (IZ) disruption; 11 eyes (91.6%) had ellipsoid zone (EZ) disruption. OCT of four affected eyes showed changes in the inner retinal structure; one eye (8.3%) had intraretinal cysts (Figure 4E); two eyes (16.7%) had cotton-wool spots and hyperreflectivity in the inner plexiform layer (IPL) (Figure 5B,D); and one eye (8.3%) had thinning of the inner nuclear layer (INL) (Figure 6D). Among the nine eyes that underwent OCTA, focal areas of flow deficit of the deep capillary plexus (DCP) corresponding to the lesions were observed in three eyes (Figure 1D,M and Figure 6B), and focal flow signal attenuation of the choriocapillaris layer corresponding to the lesions were observed in five eyes (Figure 1E,N, Figure 2E and Figure 6C,G). Flow deficit in SCP was also observed in case 6 (Figure 6A) combined with an OCT finding of INL thinning, suggesting possible concomitant PAMM.

Of the three patients (four affected eyes) who underwent mfERG examination, decreased amplitude of the P1 wave in zones 1 and 2 at the macula was observed in one eye (Figure 1T); mfERG of the contralateral eye of the same patient showed localized decreased amplitude of the P1 wave (Figure 1U) corresponding to hyporeflective lesions shown on the IR image (Figure 1P). No significant abnormalities were observed in the mfERG of the other two eyes (Figure 2J,K and Figure 6H,I).

Three patients were treated and followed up at our center. One patient received a retrobulbar injection of dexamethasone (2.5 mg) combined with racemic anisodamine (5 mg) for 3 consecutive days from the first visit. After 18 days of follow-up, the patient reported no change in their scotoma; follow-up OCT scans showed recovered ONL hyperreflectivity but persistent ONL thinning and IZ/EZ disruption (Figure 1I,R).

Three eyes received a triamcinolone (20 mg) peribulbar injection on their first visit to our center. ONL hyperreflectivity disappeared and IZ disruption partially recovered on OCT scans after 46 days of follow-up in one eye, though ONL thinning and EZ disruption persisted (Figure 2H). The other patient (with two affected eyes) reported no significant change in symptoms after 28 days of follow-up but showed improved visual acuity in both eyes with decimal VA of one eye improving from 0.3 to 0.6 with no obvious structural change on OCT scans (Figure 3F) and one eye with decimal VA improving from 0.3 to 0.8 with partially recovered IZ disruption on OCT scans (Figure 3L) during follow-up.

## 4. Discussion

AMN, first described by Bos and Deutman in 1975, is an uncommon retinal disorder characterized by reddish-brown wedge-shaped lesions in the macular area. AMN has been found to mostly affect the outer retina, especially the photoreceptor layers. Early OCT features include ONL and OPL hyperreflectivity followed by the disappearance of hyperreflectivity, thinning of ONL and IZ/EZ disruption. The exact etiology of this disease is still unclear, but it may be related to nonspecific flulike illness, fever and oral contraceptives use [2]. After the epidemic of SARS-CoV-2, many cases of AMN after SARS-CoV-2 vaccination or infection have been reported, indicating a correlation between onset of AMN and SARS-CoV-2 infection. Compared with previously reported AMN cases, the reported onset age AMN post-SARS-CoV-2 infection is older [9], and the time between AMN diagnosis and viral infection is longer [10,11]. AMN more often affected young women, consistent with the findings of this case series. In this study, one patient had a comorbidity of systemic lupus erythematosus (SLE). It has been speculated that immune-mediated vascular damage and hypercoagulability in SLE patients may aggravate the risk of outer retinal ischemia, thereby increasing the risk of AMN [12].

Previous studies reported that AMN lesions were more easily observed on IR fundus photography than on color fundus photography, appearing as dark areas with clear boundaries, especially those located outside the fovea. It should be noted that while all fundus photography images in this case series showed more notable lesions than during ophthalmoscopy, the images were obtained with an Optos 200Tx, which is based on dual-wavelength SLO technology, making the lesions more noticeable on the acquired images than during routine ophthalmoscopy. In this study, there was no obvious abnormality in the IR image of one affected eye, which may be related to the fact that it had been a long time since the patient was infected with SARS-CoV-2, and the edema of the photoreceptor cells in the outer retina had recovered enough to result in a weakened blocking effect on the reflection of the RPE layer below. It has been reported in the literature that hypofluorescence during all or late stages in FFA could be observed in approximately 21% of AMN eyes [2]. However, no abnormal findings were revealed in the FFA images of three eyes that underwent FFA examination in this study.

OCT is the most sensitive imaging examination for diagnosing AMN. OCT characteristics, including hyperreflectivity of OPL and ONL, thinning of ONL and IZ/EZ disruption, in this case series were consistent with previous reports. One patient with AMN in both eyes in this case series also showed hyperreflectivity in the retinal nerve fiber layer (RNFL) and IPL, with one eye showing INL thinning and the other eye showing intraretinal cysts, indicating that AMN post-SARS-CoV-2 infection could also affect inner retina. Invernizzi et al. reported that within 30 days from the onset of systemic symptoms, both retinal arteries and veins were larger compared to uninfected controls [13]. In the same study, SARS-CoV-2-infection-related cotton-wool spots and retinal hemorrhage were also reported [13]. Apart from AMN, RAO, RVO and acute paracentral acute middle maculopathy (PAMM) related to SARS-CoV-2 infection were also reported [8].

There have been reports that RT-PCR tests of the retinal tissue of patients who died of SARS-CoV-2 infection could detect SARS-CoV-2 virus RNA [14], though biopsy revealed no obvious retinal inflammation or venous obstruction [15]. One possible mechanism by which SARS-CoV-2 could cause microvascular dysfunction is the angiotensin-converting enzyme 2 (ACE2)-mediated entry of SARS-CoV-2 into endothelial cells, leading to endothelial cell injury and endothelialitis, thereby triggering excessive thrombin production, inhibiting fibrinolysis and activating complement pathways, initiating thromboinflammation and ultimately leading to microthrombi deposition and microvascular dysfunction [6,7]. The DCP and choriocapillaris flow signal attenuation revealed by OCTA in this study supported the theory that microvascular ischemia is involved in the pathogenesis of AMN [4].

Three patients in our case series underwent mfERG examination, revealing decreased P1 amplitude in two affected eyes but no abnormal findings in two other eyes. The low sensitivity of mfERG examination in detecting AMN may be related to the fact that most AMN lesions only affect small areas. Diminished amplitudes and diminished implicit time on the mfERG were reported in AMN eyes [2].

In this study, nine eyes underwent OCTA examination, three of which showed DCP areas of flow deficit on OCTA at the corresponding sites of the lesion and five of which showed flow signal attenuation in the choriocapillaris layer. However, it could not be ruled out that the flow signal attenuation in the choriocapillaris layers seen in this case series was caused by projection artifacts due to the overlying hyperreflective band in AMN. The involvement of the choriocapillaris layer in AMN could benefit from OCTA scans of AMN eyes long after the onset of the disease, as the outer retinal hyperreflectivity could resolve over time, and persistent choriocapillaris layer flow signal attenuation might indicate choriocapillaris involvement in AMN. In this case series, case 2 underwent OCTA imaging during follow-up visits, and persistent choriocapillaris layer flow signal attenuation could be observed even after the resolution of overlying outer retinal hyperreflectivity (Appendix A Appendix A), indicating possible choriocapillaris involvement in this case, though projection artifacts still could not be categorically ruled out. Indeed, a case series reported by Lee et al. showed that choriocapillaris flow loss could persist even after the resolution of outer retinal hyperreflectivity, indicating that the darker areas seen on OCTA more likely represented true vascular flow voids rather than artifacts [5]. Lee et al. also proposed that multilobular lesions of AMN suggested the involvement of multiple adjacent juxtafoveal choroidal lobules and that decreased flow in the choriocapillaris could be the primary insult in AMN [5]. PAMM, characterized by hyperreflectivity at the level of INL on OCT, shares overlapping features with AMN, both of which were reported to be related to DCP ischemia [4]. Thinning of the INL and flow deficits in the SCP were observed in one affected eye, and the possibility of coexisting PAMM could not be ruled out, as permanent INL thinning after the initial hyperreflective infarct is characteristic of PAMM [2,16]. PAMM combined with AMN was previously reported with or without the context of SARS-CoV-2 infection or vaccination [16,17,18]. Kulikov et al. also reported a case of AMN with chronic PAMM in the contralateral eye [16]. These findings all suggested that the two disorders might share a similar pathogenetic pathway. It should also be noted that possible flow deficit in the DCP or choriocapillaris in certain cases could not be properly detected due to artifacts caused by overlying outer retinal hyperreflectivity, as in cases 4 and 5.

Although there have been reports that AMN eyes could have persistent scotomas corresponding to EZ/IZ defects, reported outcomes of AMN have been relatively good, with decimal visual acuity of above 0.5 in most cases [2]. There is no standard treatment protocol for AMN. Hashimoto et al. reported a case that received 3 days of intravenous methylprednisolone at 1000 mg/day before switching to oral methylprednisolone at 30 mg/day, then gradually reducing the dose during the 4-month follow-up period. At follow-up, the choroidal blood flow velocity of the affected eye increased, the lesion decreased in size and the IZ/EZ recovered partially [19], though it could not be ruled out that the recovery in structure and vascular perfusion were related to the natural course of AMN itself. In this study, two patients were given retrobulbar injection of dexamethasone and racemic anisodamine for 3 consecutive days. After 18 and 45 days of follow-up, there was no reported improvement in symptoms; ONL hyperreflectivity disappeared, but hyperreflectivity in the OPL and IZ/EZ disruption persisted. One patient received peribulbar triamcinolone for both affected eyes; after follow-up, IZ disruption partially recovered compared to baseline, while visual acuity improved in the left eye despite the fact that no obvious structural change was observed on OCT scans during follow-up. These results suggested that local steroid use might contribute to recovery from AMN lesions, though it could not be ruled out that the structural and functional improvement was associated with the natural history of AMN itself.

The limitations of this study include the following: the sample size was small and only three patients had follow-up data, which could only provide limited insights for the natural disease course of AMN and its therapeutic strategy; and this study was a retrospective study and no standard examination or therapeutic protocol was applied to all enrolled patients. Large-scale, prospective studies are needed to better study the pathogenesis, natural course, intervention timing, possible treatment protocol and prognosis of AMN in the future.

## 5. Conclusions

AMN post-SARS-CoV-2 infection mostly affects young female patients and can present unilaterally or bilaterally. Dark lesions on IR reflectance and outer retinal hyperreflectivity on OCT scans are useful in diagnosing AMN. FFA images of AMN are usually unremarkable. OCTA revealed that AMN eyes could be related to DCP ischemia. OPL/ONL hyperreflectivity on OCT could disappear after short-term follow-up, but ONL thinning and IZ/EZ could persist for a long time.

## Figures and Tables

**Figure 1 diagnostics-13-03600-f001:**
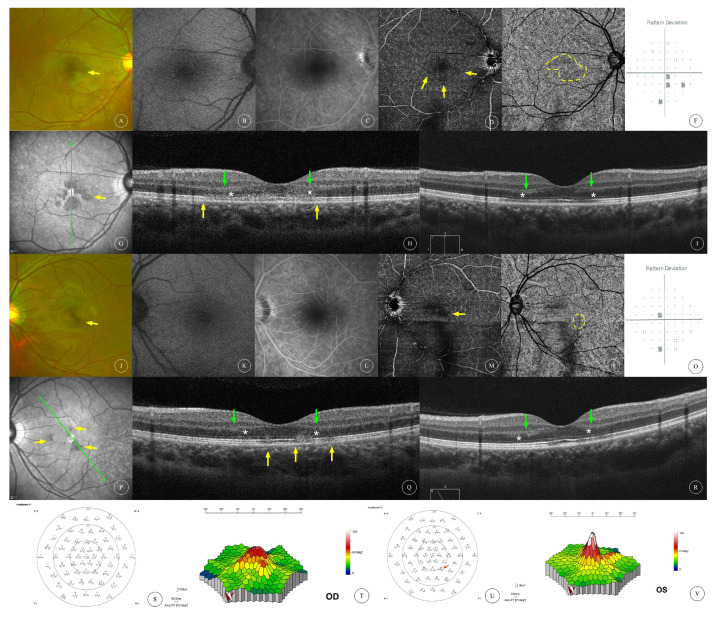
**Multimodal imaging of AMN in both eyes of case 1.** Fundus ophthalmoscopy (**A**,**J**) showed dark-red wedge-shaped lesions in the macular area of the right eye and left eye, respectively (indicated with yellow arrows). AF and FFA images ((**B**,**C**) for the right eye and (**K**,**L**) for the left eye, respectively) revealed no abnormal findings. (**D**,**M**) show the en face OCTA at DCP level of the right eye and the left eye, respectively; flow deficit at the DCP was observed (yellow arrow). (**E**,**N**) show the choriocapillaris layer of en face OCTA of the right eye and left eye, respectively; flow signal attenuation of the choriocapillaris layer at the affected area was observed (yellow circle). (**F**,**O**) show visual field (pattern deviation) of the right eye and left eye, respectively, showing paracentral scotomas in both eyes consistent with the location of the lesion. (**G**,**P**) show the IR images of the right and left eyes, respectively, revealing dark petaloid lesions around the macula (yellow arrows). (**H**,**Q**) show the OCT scans of the right and left eyes at baseline, respectively (scanning locations indicated with green arrows in (**G**,**P**), respectively). ONL hyperreflectivity (green arrows), ONL thinning (white asterisks) and IZ/EZ disruption (yellow arrows) can be seen in both eyes. The patient received retrobulbar injections of dexamethasone and racemic anisodamine injection for 3 consecutive days. (**I**,**R**) respectively, show the OCT scans after 18 days of follow-up, showing partially recovered ONL hyperreflectivity (green arrows) but persistent ONL thinning (white asterisks) and IZ/EZ disruption. (**S**,**U**) show the mfERG of the right and left eyes, respectively, showing decreased P1 amplitude in the left eye (red arrow) consistent with the location of the AMN lesion. (**T**,**V**) show the three-dimensional topographic maps of mfERG in the right and left eyes, respectively.

**Figure 2 diagnostics-13-03600-f002:**
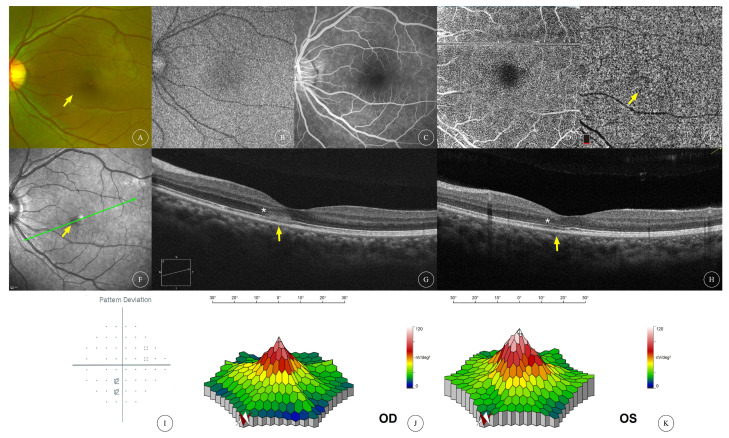
**Multimodal image of AMN in the left eye of case 2.** Fundus ophthalmoscopy (**A**) showed a dark-red lesion (yellow arrow) around the macula of the left eye at baseline. (**B**,**C**) show the AF and FFA images of the left eye, respectively, revealing no abnormal findings. No obvious flow deficit at DCP on en face OCT was observed (**D**). However, flow signal attenuation of the choriocapillaris layer was observed on en face OCTA ((**E**), yellow arrow) at the lesion site. IR revealed a focal dark area at affected area ((**F**), yellow arrow). OCT scans ((**G**), scanning location indicated with green arrow in (**F**)) revealed OPL/ONL hyperreflectivity, ONL thinning (white asterisk) and IZ/EZ disruption (yellow arrow). The patient received retrobulbar injections of dexamethasone injection 2.5 mg/0.5 mL and racemic anisodamine injection 5 mg/0.5 mL for 3 consecutive days. (**H**) shows the OCT after 46 days of follow-up, revealing that ONL hyperreflectivity disappeared but ONL thinning (white asterisk) and IZ/EZ disruption persisted (yellow arrow). (**I**) shows the visual field (pattern deviation) of the left eye, revealing a paracentral scotoma consistent with the symptom. (**J**,**K**) show the mfERG three-dimensional topographic maps of the right and left eyes, respectively, with no obvious abnormalities found.

**Figure 3 diagnostics-13-03600-f003:**
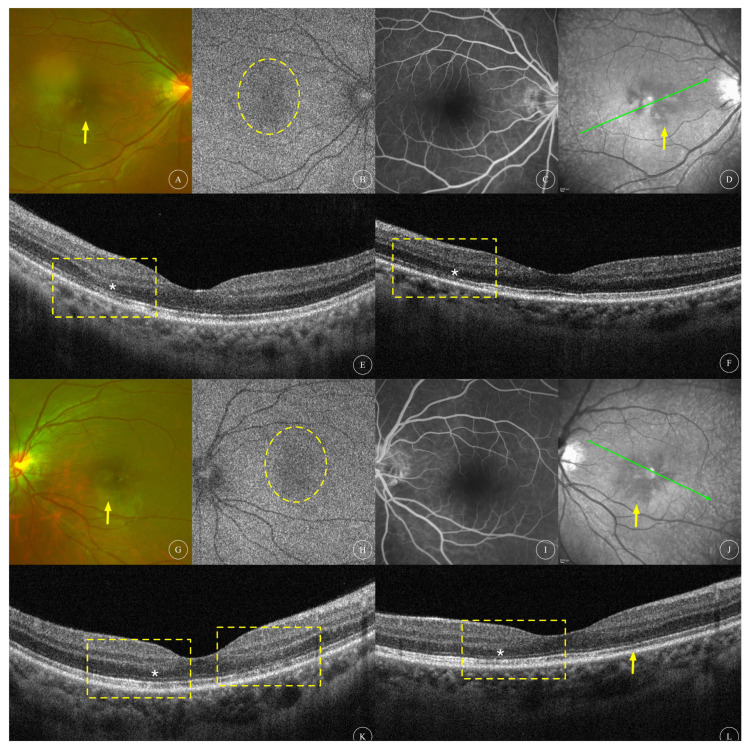
**Multimodal image of AMN in both eyes of case 3.** Fundus ophthalmoscopy (**A**,**G**) showed dark-red lesions at the macular area of the right eye and left eye, respectively (yellow arrows). Mild hypo-autofluorescence ((**B**,**H**), indicated with yellow circles) at affected area was observed in both eyes. FFA (**C**,**I**) revealed no obvious abnormal findings in both eyes. IR (**D**,**J**) revealed petaloid dark patches at affected area in both eyes (yellow arrows). OCT scans of both eyes ((**E**,**K**), scanning locations indicated with green arrows in (**D**) and (**J**), respectively) at baseline revealed ONL hyperreflectivity, ONL thinning (white asterisks) and IZ/EZ disruption in both eyes (highlighted with yellow rectangles). The patient was treated with peribulbar injection of 20 mg of triamcinolone acetonide for both eyes. OCT scans after 28 days of follow-up showed no significant change in the right eye (**F**) but partially recovered IZ structure (yellow arrow) and ONL hyperreflectivity (yellow rectangle) as well as persistent ONL thinning (white asterisk) in the left eye (**L**).

**Figure 4 diagnostics-13-03600-f004:**
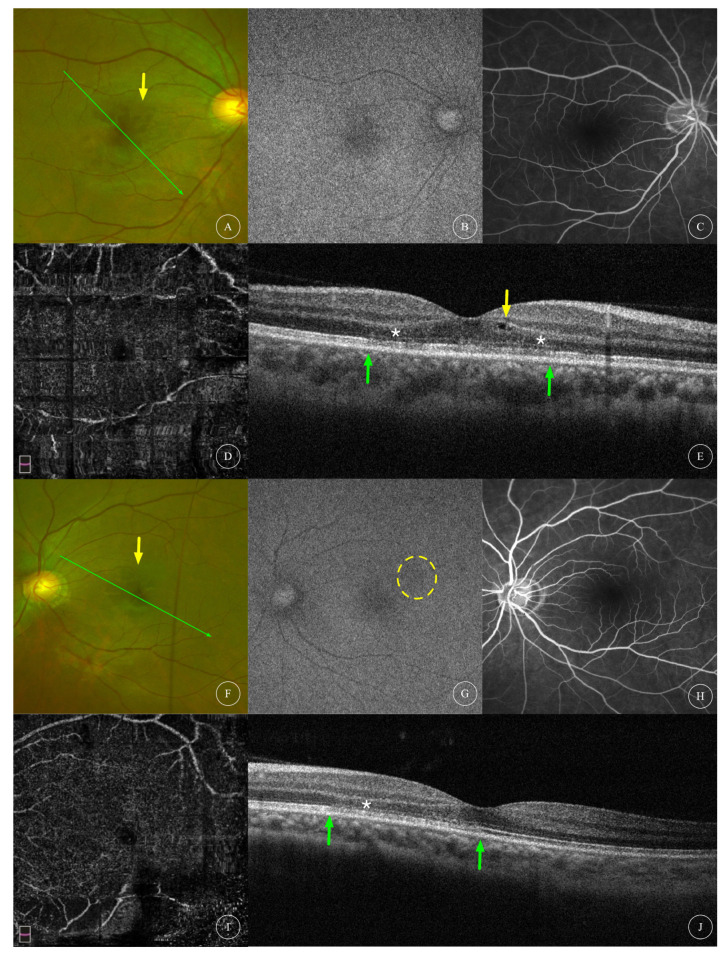
**Multimodal images of AMN of case 4.** Fundus ophthalmoscopy photographs (**A**,**F**) revealed dark-red lesions in the macular area of the right and left eyes of Case 4, respectively. (**B**,**G**) show AF images of right and left eyes of case 4, revealing no obvious abnormal finding in the right eye but a hypo-autofluorescent patch in the left eye ((**G**), yellow circle). (**C**,**H**) show FFA images of both eyes, showing neither hypoperfusion nor late-stage leakage. (**D**,**I**) show en face OCTA images of DCP; the poor quality of the images was caused by artifacts. (**E**) shows the OCT scan at the location indicated by the green arrow in (**A**), showing intraretinal cyst (yellow arrow), OPL and ONL hyperreflectivity, ONL thinning (white asterisk) and IZ/EZ disruption in the right eye (areas indicated between two green arrows). (**J**) showed the OCT scan at the location indicated by the green arrow in (**F**), showing ONL hyperreflectivity, ONL thinning (white asterisk) and IZ/EZ disruption in the left eye (areas indicated between two green arrows).

**Figure 5 diagnostics-13-03600-f005:**
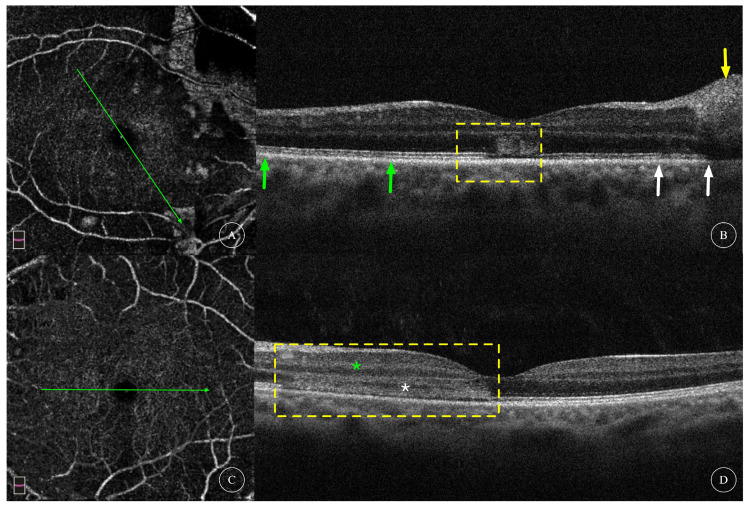
**En face OCTA and OCT B-scans of AMN of case 5.** (**A**,**C**) show DCP layer on en face of the right eye and left eye of Case 5, respectively, revealing no obvious flow deficit in both eyes. (**B**,**D**), respectively, show the OCT scans at the positions indicated by green arrows in (**A**,**C**). OCT scan of the right eye (**B**) showed ONL hyperreflectivity (yellow rectangle) and IZ/EZ disruptions (areas indicated between twin green arrows and twin white arrows) as well as hyperreflectivity of inner retina layers corresponding to a cotton-wool spot (yellow arrow). OCT scan of the left eye (**D**) showed IPL hyperreflectivity (green asterisk), OPL hyperreflectivity, ONL thinning (white asterisk), ONL hyperreflectivity and IZ/EZ disruptions (areas within yellow rectangle).

**Figure 6 diagnostics-13-03600-f006:**
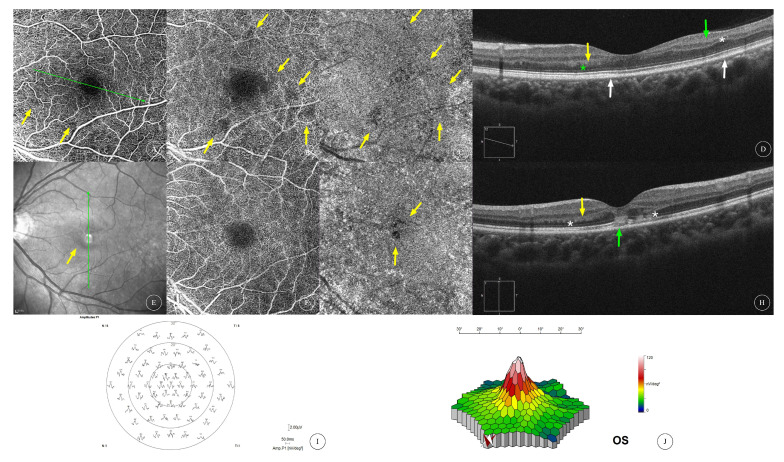
**Multimodal image of AMN of cases 6 and 7.** (**A**–**C**), respectively, show the en face OCTA images of SCP, DCP layer and choriocapillaris layer of the left eye of Case 6, showing multiple areas of flow deficits in the SCP and DCP as well as areas of flow signal attenuation of choriocapillaris layer (yellow arrows). OCT scan (**D**) at the direction indicated by the green arrow in (**A**) showed INL thinning (white asterisk), IPL hyperreflectivity (green arrow), OPL hyperreflectivity (yellow arrow), ONL thinning (green asterisk) and EZ disruption (areas between the twin white arrows). The SCP involvement as well as INL thinning suggested possible concomitant PAMM. (**E**) shows the IR image of the left eye of Case 7, showing a dark area around the macula (yellow arrow). (**F**,**G**), respectively, show the en face OCTA images of the DCP layer and choriocapillaris layer of the left eye of Case 7. No obvious flow deficit was found in the DCP, but an area of flow signal attenuation could be observed in the choriocapillaris layer. OCT scan (**H**) at the position indicated by the green arrow on (**E**) showed IPL hyperreflectivity (green arrow), OPL hyperreflectivity (yellow arrow), ONL thinning (white asterisks), ONL hyperreflectivity (green arrow) and IZ/EZ disruption involving the central fovea in the left eye of Case 7. (**I**,**J**) show the mfERG and three-dimensional topographic maps of mfERG in the left eye, respectively, with no obvious abnormalities found.

**Table 1 diagnostics-13-03600-t001:** Demographics and clinical and OCT imaging characteristics of 8 AMN patients.

Case	Sex/Age	Comorbidity	Affected Eye	Decimal BCVA	Symptoms	Time from Fever to Onset of Symptoms	OCT Characteristics					
							OPL Hyperreflectivity	ONL Hyperreflectivity	ONL Thinning	IZ Disruption	EZ Disruption	Other OCT Characteristics
**1**	M/27	None	Bilateral	1.0 1.0	Scotoma, photopsia	2	− −	+ +	+ +	− −	+ +	
**2**	F/25	None	Left eye	1.0	Scotoma, photopsia	2	−	+	+	+	+	
**3**	F/27	None	Bilateral	0.3 0.3	Scotoma, decreased vision	2	− −	− −	+ +	+ +	+ +	
**4**	F/29	None	Bilateral	0.12 0.4	Scotoma, decreased vision	10	+ −	+ +	+ +	+ +	+ +	Intraretinal cysts in the right eye
**5**	F/29	Hyperthyroidism	Bilateral	1.0 1.0	Scotoma, decreased vision	1	− +	+ +	+ +	+ +	+ +	Bilateral cotton-wool spots and IPL hyperreflectivity
**6**	M/32	None	Left eye	1.0	Scotoma	2	+	−	+	−	+	INL thinning
**7**	F/16	None	Left eye	0.5	Scotoma	2	+	+	+	+	+	
**8**	F/54	SLE	Left eye	0.80	None reported *	N/A	+	−	+	−	−	

OCT, optical coherence tomography; BCVA, best-corrected visual acuity; OPL, outer plexiform layer; ONL, outer nuclear layer; IZ, interdigitation zone; EZ, ellipsoid zone; IPL, inner plexiform layer; SLE, systemic lupus erythematosus. In the OCT characteristics columns, for patients that were affected bilaterally, the first row describes the right eye while the second row describes the left eye; “+” means that the OCT scans showed corresponding features, and “−” means that the OCT scans showed no corresponding features. * The eighth patient was asymptomatic and presented to the clinic 41 days after the initial fever.

## Data Availability

The data presented in this study are available on request from the corresponding author. The data are not publicly available due to potentially privacy-compromising information regarding the research participants.

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
