# Peer review of "Acute Macular Neuroretinopathy after SARS-CoV-2 Infection: An Analysis of Clinical and Multimodal Imaging Characteristics"

_diagnostics, 2023, doi:10.3390/diagnostics13243600_

Round 1

Reviewer 1 Report

Comments and Suggestions for Authors

Many thanks to the editorial for an opportunity to review this paper. Dr Tang and colleagues describe a series of AMN cases in COVID patients. The pathophysiology and origin of AMN is interesting topic, however the paper requires significant modification before further consideration.

Abstract:

Line 16: please provide more information on the methods section

Line 23-24: it senssless to say about particular cases here

Line 26: conclusion – please make your conclusion based on the results section be. For example, nothing was said about laterality but then this fact unexpectedly appears in the conclusion.

Line 64: “Infrared (IR) fundus photography” -> IR reflectance

Line 85: Since AMN may be related to various concomitant medication, please clarify if the patients have been receiving any drugs for COVID infection (NSAID, antiviral, glucocorticoids, anticoagulants, etc.)

Line 89: “Fundus examination of all affected” - Please note that pictures you provide are obtained with Optos which is based on dual wavelength SLO technology, that’s why appearance of the lesions is more notable than during routine ophthalmoscopy, where they are typically poorly distinguishable

Figure 1: choriocapillaris hypoperfusion which, as you stated, indicated by the arrows in 1E and 1N is hardly visible, if any. That’s not against choriocapillaris involvement in AMN, which might be just not severe enough to cause flow signal attenuation… but please be accurate in the marking the figures. Same for figure 2. Please also keep in mind that outer retinal hyperreflectivity, when significant, leads to the weakening of the flow signal how it is seen in figure 4. In total, I recommend avoiding interpretation of choriocapillaris changes in acute AMN. This is also supported by rarely noticeable choriocapillaris alteration after resolution of acute AMN.

Please indicate AMN lesions on your cross-sectional scans in all figures

Figure 3: please provide mfERG… it is not clear why BCVA is so poor in this case with relatively preserved retina

Figure 4F not clear what exactly the arrow indicates… CWS I suppose?

Figure 4: I recommend to divide this figure for cases 4 and 5. I current form it may mislead the readers

Classical description of AMN by OCT includes ONL hyperreflectivity, rarely hyperreflectivity at the border between ONL and OPL, and only a few studies describe IPL hyperreflectivity (the last, I personally believe is not fully correct) (Bhavsar KV et al. Acute macular neuroretinopathy: A comprehensive review of the literature. Surv Ophthalmol. 2016 Sep-Oct;61(5):538-65). It is also seen from your cases which demonstrate ONL hyperreflectivity and hyperreflectivity at the border between ONL and OPL, but almost no OPL hyperreflectivity (I was able to distinguish it only in 4B).  Otherwise please clearly indicate on the figures where OPL hyperreflectivity takes place.

260 please fist explain to the readers that INL thinning is a characteristic feature of resolved PAMM lesion

261 please discuss previously reported combinations of AMN and resolved PAMM regardless of COVID (Kulikov AN, Maltsev DS, Leongardt TA. RETINAL MICROVASCULATURE ALTERATION IN PARACENTRAL ACUTE MIDDLE MACULOPATHY AND ACUTE MACULAR NEURORETINOPATHY: A QUANTITATIVE OPTICAL COHERENCE TOMOGRAPHY ANGIOGRAPHY STUDY. Retin Cases Brief Rep. 2020 Fall;14(4):343-351.)

Although you consider choriocapillaris hypoperfusion in your cases, in the introduction and discussion AMN is described as a condition caused solely by DCP ischemia.

Line 292: “hyper-reflective lesions on OCT scans are useful” I do not think that the lesions are useful. That’s may by correct for OCT, as you said for IR, but definitely not for the lesions…

Author Response

Thank you for taking the time to review this manuscript.The point-by-point response to the reviewer's comments is the following:

Response to Reviewer 1 Comment:

Thank you very much for taking the time to review this manuscript. Please find the detailed responses below and the corresponding revisions highlighted in the re-submitted files.

Abstract:

Line 16: please provide more information on the methods section

More information has been added to the methods section of the abstract, though due to the word limit on the abstract only selected details of study methods was included in the section.

Line 23-24: it senssless to say about particular cases here

Results regarding particular cases were reworded and re-structured to better fit the content of the Results section.

Line 26: conclusion – please make your conclusion based on the results section be. For example, nothing was said about laterality but then this fact unexpectedly appears in the conclusion.

Details have been added to Methods and Results sections to better clarify the conclusion drawn in the abstract.

Line 64: “Infrared (IR) fundus photography” -> IR reflectance

We have made the change as suggested, thank you for pointing this out.

Line 85: Since AMN may be related to various concomitant medication, please clarify if the patients have been receiving any drugs for COVID infection (NSAID, antiviral, glucocorticoids, anticoagulants, etc.)

Details have been added to the text. All patients received NSAIDs for COVID-19 induced fever and none reported use of an-ticoagulants. All patients denied any history of oral contraceptives use. All but one patient (case 8) denied use of glucocorticoids. Case 8 received regular oral glucocorticoids for her comorbidity of systemic lupus erythematosus (SLE). (line 95-98)

Line 89: “Fundus examination of all affected” – Please note that pictures you provide are obtained with Optos which is based on dual wavelength SLO technology, that’s why appearance of the lesions is more notable than during routine ophthalmoscopy, where they are typically poorly distinguishable

Thank you for pointing this out, we have added to the discussion section to point out the reason fundus photographs in this study all showed clearly demarcated lesions might have benefited from the dual wavelength SLO technology of the Optos 200Tx we used in our study. (line 253-256)

Figure 1: choriocapillaris hypoperfusion which, as you stated, indicated by the arrows in 1E and 1N is hardly visible, if any. That’s not against choriocapillaris involvement in AMN, which might be just not severe enough to cause flow signal attenuation… but please be accurate in the marking the figures. Same for figure 2. Please also keep in mind that outer retinal hyperreflectivity, when significant, leads to the weakening of the flow signal how it is seen in figure 4. In total, I recommend avoiding interpretation of choriocapillaris changes in acute AMN. This is also supported by rarely noticeable choriocapillaris alteration after resolution of acute AMN.

Please indicate AMN lesions on your cross-sectional scans in all figures

A paragraph regarding flow signal attenuation on CC layer in AMN was added to the manuscript. It could not be ruled out that the flow signal attenuation in the choriocapillaris layers seen in this case series was caused by projection artifact due to the overlying hyperreflective band in AMN. The involvement of choriocapillaris layer in AMN could benefit from OCTA scans of AMN eyes long after the onset of the disease, as the outer retinal hyper-reflectivity could resolve over time and persistent choriocapillaris layer flow signal attenuation might indicate choriocapillaris involvement in AMN. Previously a case series reported by Lee et al. showed that choriocapillaris flow loss could persist even after resolution of outer retinal hyperreflectivity, indicating that the darker areas seen on OCTA more likely represented true vascular flow void rather than artifacts. Lee et al. also proposed that multilobular lesions of AMN suggested involvement of multiple, adjacent juxtafoveal choroidal lobules and that decreased flow in choriocapillaris could be the primary insult in AMN[.

All figures have been updated with clearer indicators.

Figure 3: please provide mfERG… it is not clear why BCVA is so poor in this case with relatively preserved retina

Unfortunately this patient did not undergo mfERG test, though the central fovea involvement and IZ/EZ disruption may have contributed to the poor BCVA in this case.

Figure 4F not clear what exactly the arrow indicates… CWS I suppose?

This yellow arrow was indeed used to indicate CWS in this particular case. Figure 4 has been updated with better-positioned indicators.

Figure 4: I recommend to divide this figure for cases 4 and 5. I current form it may mislead the readers

Images for case four and five have been re-uploaded as separated figures (Fig 4 and Fig 5) as suggested.

Classical description of AMN by OCT includes ONL hyperreflectivity, rarely hyperreflectivity at the border between ONL and OPL, and only a few studies describe IPL hyperreflectivity (the last, I personally believe is not fully correct) (Bhavsar KV et al. Acute macular neuroretinopathy: A comprehensive review of the literature. Surv Ophthalmol. 2016 Sep-Oct;61(5):538-65). It is also seen from your cases which demonstrate ONL hyperreflectivity and hyperreflectivity at the border between ONL and OPL, but almost no OPL hyperreflectivity (I was able to distinguish it only in 4B).  Otherwise please clearly indicate on the figures where OPL hyperreflectivity takes place.

We reviewed all OCT B-scan images and re-evaluated the outer retinal hyperreflectivity in all cases. When the hyperreflectivity was positioned at ONL or at the border between OPL and ONL, this was classified as ONL hyperreflectivity. When obvious OPL hyperreflectivity could be observed on OCT B-scans, this was classified as OPL hyperreflectivity. All image legends and table information have been updated accordingly.

260 please fist explain to the readers that INL thinning is a characteristic feature of resolved PAMM lesion

We’ve added the relevant discussion regarding INL thinning as a characteristic feature of resolved PAMM in the manuscript as suggested. We’ve also added en face OCTA of SCP slab of the same eye (which revealed areas of flow deficit) in Figure 6 to support the theory of possible concomitant PAMM.

261 please discuss previously reported combinations of AMN and resolved PAMM regardless of COVID (Kulikov AN, Maltsev DS, Leongardt TA. RETINAL MICROVASCULATURE ALTERATION IN PARACENTRAL ACUTE MIDDLE MACULOPATHY AND ACUTE MACULAR NEURORETINOPATHY: A QUANTITATIVE OPTICAL COHERENCE TOMOGRAPHY ANGIOGRAPHY STUDY. Retin Cases Brief Rep. 2020 Fall;14(4):343-351.)

Relevant discussion regarding INL thinning in resolved PAMM, concomitant AMN/PAMM and PAMM in the contralateral eye of AMN affected eye has been added to the discussion section as suggested (line 307-318).

Although you consider choriocapillaris hypoperfusion in your cases, in the introduction and discussion AMN is described as a condition caused solely by DCP ischemia.

We have rectified this and added possible choriocapillaris involvement as previous reports suggested in the introduction and discussion (line 292-307).

Line 292: “hyper-reflective lesions on OCT scans are useful” I do not think that the lesions are useful. That’s may by correct for OCT, as you said for IR, but definitely not for the lesions…

Thank you for pointing this out, we have reworded this sentence to “Dark lesions on IR reflectance and outer retinal hyperreflectivity on OCT scans are useful in diagnosing AMN” so that this sentence could make more sense. We mean to say that these imaging characteristics are useful in diagnosing AMN in clinical settings.

Reviewer 2 Report

Comments and Suggestions for Authors

Interesting study. Need English editing...

Comments on the Quality of English Language

Poor.

Need editing.

For example ;

In introduction part. Last paragraph has  to be rewritten.

Author Response

Thank you very much for taking the time to review this manuscript. The manuscript in its entirety has been reviewed and edited. Corresponding changes to the wording and certain technical terms as suggested by other reviewers have been made to improve the general coherency of the manuscript. I hope the updated manuscript is more fluent and eloquent to its readers.

Reviewer 3 Report

Comments and Suggestions for Authors

Overview and strengths:                                  

Overview: Overall, this case series provides interesting preliminary data on acute macular neuroretinopathy (AMN) associated with COVID-19 infection. The imaging and clinical descriptions are detailed.

Strengths:

·      The topic is novel and clinically relevant given the pandemic. Reports of COVID-related retinal findings are important to advance understanding.

·      The cases add to the limited literature on AMN associated with COVID-19. Documentation of clinical features through multimodal imaging is a strength.

·      The authors propose a reasonable hypothesis connecting AMN to vascular damage from COVID-19, based on current knowledge. However, this remains speculative without more evidence.

·      Multimodal imaging with OCT, OCTA, FA, IR, etc. allows thorough phenotypic characterization of the AMN lesions. This is a major strength of the paper.

·      The OCT findings are analyzed systematically and described in detail, which provides useful information on the retinal layers and structures affected in COVID-related AMN.

Major issues:

Materials and Methods:

·      More details are needed on the study design, sampling method, inclusion/exclusion criteria, etc. Was this a retrospective case series? How were patients identified and selected? This is important for assessing potential selection bias.

Results:

·      Please add arrows to all figures to indicate pathology when it’s present. Some figures have yellow arrows to indicate this, while others don’t. For consistency and for ophthalmology trainees with less expertise in retinal imaging analysis, I believe this is an important point to address.

·      For all the cases where a flow deficit is noted, (e.g., Figures 1E and 1N) I need to see the corresponding B-scan with flow overlay to make sure that what we’re saying is indeed a flow deficit rather than an OCTA artifact. This is especially true for images that are showing motion artifact such as Figure 1N. I believe this is a crucial point since hypoperfusion seems to be tied to the pathology in AMN, and with little literature on SARS-CoV-2 infection induced AMN, and the low number of patients in this report, this finding should be verified first. Note that the authors don’t have to include the corresponding B-scan with flow overlay in the manuscript, but can upload the them in a supplementary file for the reviewers to double-check and can choose to include them (or not) in the published version as supplementary material.

Minor issues:

Overall:

·      The authors use the term ‘flow defect’ while the more commonly used term is ‘flow deficit’ which describes areas on OCTA images where there is lack of blood flow. If this is what you were referring to, please consider replacing all the terms with the commonly used term ‘deficit’.

Introduction:

·      Page 2, line 53: Please remove “after 2019-nCoV infection in our center” as it seems to be redundant with the sentence preceding it.

·      Page 2, line 38: "oral contraceptive" should probably be "oral contraceptives" if referring to the class of medications in general.

·      Page 2, line 40: Change "The etiology of AMN is still unclear." to "The exact etiology of AMN remains unclear."

Materials and Methods:

·      Page 2, line 57: the verb tense should match the subject, so it should be "were retrospectively reviewed" instead of "was retrospectively reviewed" since you're talking about records of multiple patients.

·      Page 2, line 64: "Infrared (IR) fundus photography were performed" should be "Infrared (IR) fundus photography was performed".

Results:

·      Page 3, line 84: ‘presented no symptoms’ should be ‘presented with no symptoms.’

·      Page 7, line 74: There is a minor typographical error "wer observed" which should be "were observed".

·      Table 1: in its current format, it is confusing to determine in the ‘Other OCT characteristics’ column if the intraretinal cysts in the right eye and the bilateral spots and IPL hyper-reflectivity belong to patient #4 or patient #4 and patient #5, respectively. Please modify the format to clarify this for the readers.

·      Figure 2D and 2E: these are en face OCTA images and not en face OCT images. Please correct the figure legend.

·      Figures 4C: you indicate that these are DCP slabs. Yet, I see prominent superficial vessels. Please double-check that there is no segmentation error in this figure. If there is one, either correct the segmentation manually on the OCTA device and reupload a better OCTA image or choose another image if available. If this is not possible, please indicate in one sentence the poor quality of the image. In other words, since the segmentation is not perfect, there might be areas of non-perfusion that we cannot detect due to the poor quality of the image.

·      Figure 4D: the yellow arrow seems misplaced. Please remove it or indicate in the figure legend what it is referring to.

·      Did the 3 patients who receive follow-up undergo OCTA imaging? If so, it is important to see if there were any changes noted in terms of flow deficits.

Discussion:

·      Terminology Consistency: "new coronavirus" appears a few times. To ensure uniformity, replace "new coronavirus" with "SARS-CoV-2" throughout.

·      Page 9, line 210: Refine the phrase "after the epidemic of new coronavirus infection" as such "following the SARS-CoV-2 pandemic".

·      Page 9, line 258: there’s a typo ‘th.ree’ should be ‘three’.

Comments on the Quality of English Language

The manuscript could benefit from professional English editing. There are several instances in the text where both grammar and sentence structure could be improved for better flow of ideas and readability. This includes the figure legends as well (e.g. Figure 4 legend, ‘at the located’, ‘intraretinal cystic’…).

Author Response

Many thanks for taking the time to review the manuscript. The point-by-point response is the following:

Response to Reviewer 3

Thank you very much for taking the time to review this manuscript. Please find the detailed responses below and the corresponding revisions highlighted in the re-submitted files.

Major issues:

Materials and Methods:

  • More details are needed on the study design, sampling method, inclusion/exclusion criteria, etc. Was this a retrospective case series? How were patients identified and selected? This is important for assessing potential selection bias.

More details have been added to the Materials and Methods section. This was indeed a retrospective observational study. Further details about inclusion and exclusion criteria, as well as diagnostic criteria were also included in the updated manuscript (linen 79-86).

Results:

  • Please add arrows to all figures to indicate pathology when it’s present. Some figures have yellow arrows to indicate this, while others don’t. For consistency and for ophthalmology trainees with less expertise in retinal imaging analysis, I believe this is an important point to address.
  • For all the cases where a flow deficit is noted, (e.g., Figures 1E and 1N) I need to see the corresponding B-scan with flow overlay to make sure that what we’re saying is indeed a flow deficit rather than an OCTA artifact. This is especially true for images that are showing motion artifact such as Figure 1N. I believe this is a crucial point since hypoperfusion seems to be tied to the pathology in AMN, and with little literature on SARS-CoV-2 infection induced AMN, and the low number of patients in this report, this finding should be verified first. Note that the authors don’t have to include the corresponding B-scan with flow overlay in the manuscript, but can upload the them in a supplementary file for the reviewers to double-check and can choose to include them (or not) in the published version as supplementary material.

Flow signal attenuation was used instead when referring to choriocapillaris as it cannot be categorically ruled out that it was caused by artifact. Relevant discussion regarding flow signal attenuation on choriocapillaris in AMN was added to the manuscript. As previously reported by Lee et al., choriocapillaris flow loss could persist even after resolution of outer retinal hyperreflectivity, indicating that the darker areas seen on OCTA more likely represented true vascular flow void rather than artifacts. Unfortunately only one case in our case series with follow-up data underwent OCTA examination at follow-up visits, though in this case persistent flow signal attenuation on CC despite resolved overlying outer retinal hyperreflectivity has also been observed (supplementary figure 3), suggesting possible CC involvement in this particular case.

Supplementary figure 1 and 2 have been uploaded to include the corresponding B-Scans with flow overlay for cases with en face OCTA images. Updated figures with indicators have be re-uploaded as suggested.

Minor issues:

Overall:

  • The authors use the term ‘flow defect’ while the more commonly used term is ‘flow deficit’ which describes areas on OCTA images where there is lack of blood flow. If this is what you were referring to, please consider replacing all the terms with the commonly used term ‘deficit’.

Thank you for pointing this out, changes on the wording have been made as suggested.

Introduction:

  • Page 2, line 53: Please remove “after 2019-nCoV infection in our center” as it seems to be redundant with the sentence preceding it.
  • Page 2, line 38: "oral contraceptive" should probably be "oral contraceptives" if referring to the class of medications in general.
  • Page 2, line 40: Change "The etiology of AMN is still unclear." to "The exact etiology of AMN remains unclear."

Changes on wording have been made as suggested.

Materials and Methods:

  • Page 2, line 57: the verb tense should match the subject, so it should be "were retrospectively reviewed" instead of "was retrospectively reviewed" since you're talking about records of multiple patients.
  • Page 2, line 64: "Infrared (IR) fundus photography were performed" should be "Infrared (IR) fundus photography was performed".

 Changes on wording have been made as suggested.

Results:

  • Page 3, line 84: ‘presented no symptoms’ should be ‘presented with no symptoms.’
  • Page 7, line 74: There is a minor typographical error "wer observed" which should be "were observed".

Changes on wording have been made as suggested.

  • Table 1: in its current format, it is confusing to determine in the ‘Other OCT characteristics’ column if the intraretinal cysts in the right eye and the bilateral spots and IPL hyper-reflectivity belong to patient #4 or patient #4 and patient #5, respectively. Please modify the format to clarify this for the readers.

Row height has been adjusted to clarify which case the “Other OCT characteristics” indicated.

  • Figure 2D and 2E: these are en face OCTA images and not en face OCT images. Please correct the figure legend.

The figure legend has been corrected as suggested.

  • Figures 4C: you indicate that these are DCP slabs. Yet, I see prominent superficial vessels. Please double-check that there is no segmentation error in this figure. If there is one, either correct the segmentation manually on the OCTA device and reupload a better OCTA image or choose another image if available. If this is not possible, please indicate in one sentence the poor quality of the image. In other words, since the segmentation is not perfect, there might be areas of non-perfusion that we cannot detect due to the poor quality of the image.

A correct segmented DCP slab has been re-uploaded in place. Though the poor quality of the image did prevent us from making any meaningful interpretations regarding DCP or CC involvement in case 4 and 5, for which we have added a sentence in discussion pointing this out.

  • Figure 4D: the yellow arrow seems misplaced. Please remove it or indicate in the figure legend what it is referring to.

The yellow arrow was used to indicate the cotton-wool spot, we have uploaded all figures with better-positioned indicators.

  • Did the 3 patients who receive follow-up undergo OCTA imaging? If so, it is important to see if there were any changes noted in terms of flow deficits.

Unfortunately only one case in our case series with follow-up data underwent OCTA examination at follow-up visits, though in this case persistent flow signal attenuation on CC despite resolved overlying outer retinal hyperreflectivity has also been observed (supplementary figure 3), suggesting possible CC involvement in this particular case. We have added the relevant discussion at line 298-307.

Discussion:

  • Terminology Consistency: "new coronavirus" appears a few times. To ensure uniformity, replace "new coronavirus" with "SARS-CoV-2" throughout.
  • Page 9, line 210: Refine the phrase "after the epidemic of new coronavirus infection" as such "following the SARS-CoV-2 pandemic".
  • Page 9, line 258: there’s a typo ‘th.ree’ should be ‘three’.

 Changes on wording have been made as suggested.

Round 2

Reviewer 1 Report

Comments and Suggestions for Authors

Thanks to the authors. All questions were answered.